# High Seroprevalence of SARS-CoV-2 IgG and RNA among Asymptomatic Blood Donors in Makkah Region, Saudi Arabia

**DOI:** 10.3390/vaccines10081279

**Published:** 2022-08-08

**Authors:** Kamal H. Alzabeedi, Raafat T. M. Makhlof, Rowaida A. Bakri, Ashraf A. Ewis, Heba W. Alhamdi, Turki M. A. Habeebullah, Asim A. Khogeer, Eman A. A. Mulla, Safiah A. M. Roshan, Fadel H. Qabbani, Fayez H. Hafez, Rehab G. Alqurashi, Muhammad O. Babalghaith, Ahmad A. Ghouth, Mohammed H. Alhazmi, Othman M. Fallatah, Saeed A. Badahdah, Duaa I. A. Endergiri, Boshra M. Albarakati, Sayed F. Abdelwahab

**Affiliations:** 1Departments of Medical Research/Clinical Biochemistry, The Regional Laboratory, P.O. Box 55028, Makkah 21955, Saudi Arabia; 2Department of Parasitology, Faculty of Medicine, Umm Al Qura University, P.O. Box 715, Makkah 21955, Saudi Arabia; 3Department of Parasitology, Faculty of Medicine, Minia University, Minia 61511, Egypt; 4Department of Public Health, College of Health Sciences-AlQunfudah, Umm Al Qura University, Al Qunfudah 28821, Saudi Arabia; 5Department of Biology, College of Sciences, King Khalid University, P.O. Box 960, Abha 61413, Saudi Arabia; 6The Custodian of the Two Holy Mosques Institute for Hajj Research, Umm Al Qura University, P.O. Box. 6287, Makkah 21955, Saudi Arabia; 7Research Department, The Strategic Planning, General Directorate of Health Affairs Makkah Region, Ministry of Health, P.O. Box 6251, Makkah 21955, Saudi Arabia; 8Medical Genetics Unit, Maternity & Children Hospital, Makkah Healthcare Cluster, Ministry of Health, P.O. Box 6251, Makkah 21955, Saudi Arabia; 9Departments of Immunology, The Regional Laboratory, P.O. Box 55028, Makkah 21955, Saudi Arabia; 10Departments of Serology, The Regional Laboratory, P.O. Box 55028, Makkah 21955, Saudi Arabia; 11Departments of TB, The Regional Laboratory, P.O. Box 55028, Makkah 21955, Saudi Arabia; 12Central Blood Bank, The Regional Laboratory, P.O. Box 55028, Makkah 21955, Saudi Arabia; 13Laboratory and Blood Bank, Al Noor Specialist Hospital, Makkah 21955, Saudi Arabia; 14Department of Pharmaceutics and Industrial Pharmacy, College of Pharmacy, Taif University, P.O. Box 11099, Taif 21944, Saudi Arabia

**Keywords:** blood donors, COVID-19, IgG, SARS-CoV-2, seroprevalence

## Abstract

The gold-standard approach for diagnosing and confirming Severe Acute Respiratory Syndrome Corona Virus-2 (SARS-CoV-2) infection is reverse transcription-polymerase chain reaction (RT-PCR). This method, however, is inefficient in detecting previous or dormant viral infections. The presence of antigen-specific antibodies is the fingerprint and cardinal sign for diagnosis and determination of exposure to infectious agents including Corona virus disease-2019 (COVID-19). This cross-sectional study examined the presence of SARS-CoV-2 spike-specific immunoglobulin G (IgG) among asymptomatic blood donors in Makkah region. A total of 4368 asymptomatic blood donors were enrolled. They were screened for spike-specific IgG using ELISA and COVID-19 RNA by real-time PCR. COVID-19 IgG was detected among 2248 subjects (51.5%) while COVID-19-RNA was detected among 473 (10.8%) subjects. The IgG frequency was significantly higher among males and non-Saudi residents (*p* < 0.001 each) with no significant variation in IgG positivity among blood donors with different blood groups. In addition, COVID-19 RNA frequency was significantly higher among donors below 40-years old (*p* = 0.047, χ^2^ = 3.95), and non-Saudi residents (*p* = 0.001, χ^2^ = 304.5). The COVID-19 IgG levels were significantly higher among the RNA-positive donors (*p* = 001), and non-Saudi residents (*p* = 0.041), with no variations with age or blood group (*p* > 0.05). This study reveals a very high prevalence of COVID-19 IgG and RNA among asymptomatic blood donors in Makkah, Saudi Arabia indicating a high exposure rate of the general population to COVID-19; particularly foreign residents. It sheds light on the spread on COVID-19 among apparently healthy individuals at the beginning of the pandemic and could help in designing various control measures to minimize viral spread.

## 1. Introduction

Severe pneumonia cases with an unknown source were first reported in Wuhan, China, in late 2019, before spreading across the country and then around the world. The pathogen responsible for this illness was identified as Severe Acute Respiratory Syndrome Corona Virus-2 (SARS-CoV-2), a novel beta coronavirus, and the sickness was termed corona virus disease 2019 (COVID-19). Within months, COVID-19 cases had been reported in almost every continent in the world [1]. On 2nd March 2020, a case of SARS-CoV-2 was detected in a Saudi Arabian man traveling from Iran [2], triggering the implementation of many protective measures, including partial and then total lockdown. At that time, the Saudi Ministry of Health (MOH) got highly alerted, and the first diagnosed patient and all his contacts were quarantined. The virus’s subsequent local spread, on the other hand, was rapid, with many instances being discovered in the same location. The earliest instances were linked to residents returning from Iran, who most likely took the virus back with them, whereas later cases were related to local community transmission activities.

Fortunately, the kingdom has only experienced one peak so far, from June to August 2020, owing to the Saudi health authority’s excellent response and efforts since the beginning of the epidemic, and the lockdown was lifted on 21 June 2020. As of 21st February 2021, there were 373,702 confirmed SARS-CoV-2 cases in the Kingdom with 6,445 deaths [3]. As of 17 March 2022, there has been 464 million confirmed SARS-CoV-2 cases worldwide, with 6,082,852 deaths (http://www.worldometers.info/coronavirus/, accessed on 17 March 2022). In Saudi Arabia, there were 749,268 confirmed COVID-19 cases, including 731,638 recovered cases, 8606 active cases, and 9024 deaths. These results indicated a recovery rate of over 98% and a case fatality rate of 0.39% (https://covid19.moh.gov.sa/, accessed on 17 March 2022). On the other hand, according to a multicenter investigation in Saudi Arabia, the virus had a median incubation period of six days. The most common symptoms were cough (89.4%), fever (85.6%), and sore throat (81.6%), with 20.1 % of the infected patients suffering from underlying comorbidities, e.g., diabetes, hypertension, asthma, etc. [2].

The reverse transcription-polymerase chain reaction (RT-PCR) is the gold-standard method for detecting and confirming SARSCoV-2 infection [4]. However, this approach is incapable of detecting past or dormant viral infections such as Hepatitis B or herpes viruses. As a result, serological tests could be a useful tool for identifying the prevalence of previous SARS-CoV-2 infections in the general population. This would help with current efforts to minimize viral transmission rates and the creation of comprehensive public controlling and preventive health strategies.

Immune reactivity to SARS-CoV-2 varies and is necessary for efficient elimination of the infection. During SARS-CoV-2 infection, immunoglobulin (Ig) M and IgG are generated at various times. IgG emerges after IgM probably by the end of the first week of infection and can last for months, if not years [5]. In terms of SARS-CoV-2 IgG, multiple investigations have shown that most infected cases had seroconverted by the second- or third-week following infection. IgG levels peaked at roughly 1–2 months and continued for up to 4–5 months in a subset of individuals as reported [6,7,8]. Seropositivity is higher in patients with symptomatic SARS-CoV-2 infection than in those who are asymptomatic [9]. Furthermore, increased seropositivity was linked to the severity of the SARS-CoV-2 infection [10]. Despite their limitations in estimating the incidence of the SARS-CoV-2 pandemic, serological testing can be an important tool in monitoring the disease’s spread, if performed regularly and serially. Since the start of the pandemic, Saudi Arabia has performed four SARS-CoV-2 IgG investigations [11,12,13,14]. Three of these studies were among blood donors during the early stages of the pandemic (January to May, May, and June 2020) [12,13,14], while one looked at healthcare workers (HCWs) in May 2020 [11]. The data demonstrated a wide range of SARS-CoV-2 IgG positivity, ranging from 0% to 19.3 %. In May 2020, reports from Saudi Arabia looked at seroprevalence and discovered that it was 0% [13] and 1.4 [14] among blood donors and 2.36% among HCWs [11]. Another study performed at Al-Madinah Al-Monawarah region showed 19.3% prevalence among 1211 healthy blood donors [12]. In addition, a community-based study in Jazan Province revealed a 26% SARS-CoV-2 IgG seroprevalence among 594 participants [15]. However, no one has looked at SARS-CoV-2 seroprevalence to better explain the disease’s transmission among asymptomatic blood donors in Makkah, Saudi Arabia after the peak and before the vaccine was introduced.

Several host and viral factors contribute to the severity of COVID-19 infections. In this regard, previous investigations have revealed that the ABO blood group classification, which is commonly employed in clinical practice, confers differential viral susceptibility and disease severity caused by viruses, including SARS-CoV-1 [16]. Indeed, as with SARS-CoV-1, blood groups can play a direct role in infection by acting as virus receptors or co-receptors. Preliminary evidence revealed a link between blood group antigens and higher susceptibility to or severity of COVID-19 disease [17,18,19,20]. Individuals with blood group A had a greater incidence of severe COVID-19 symptoms [21], whereas blood group O carriage was a protective factor. 

In this study, we measured the levels of IgG antibodies targeting the SARS-CoV-2 spike protein among asymptomatic blood donors in Makkah region of Saudi Arabia after the peak of the COVID-19 pandemic. ABO blood types were used to determine the distribution of IgG-positive cases. The current research sheds light on the impact of SARS-CoV-2 infection in Saudi Arabia’s Makkah region.

## 2. Subjects and Methods

### 2.1. Study Settings

The current study is a cross sectional study that included 4368 blood donors at Makkah’s Central Blood Bank, which is the region’s major blood bank and supplier of almost all governmental and private hospitals in the region. It is one of the Saudi Arabia’s best blood banks, being certified and accredited by the Saudi Central Board for Accreditation of Healthcare Institutions (CBAHI), which is the official institution in charge of granting accreditation certificates for all government and private healthcare facilities in the country.

### 2.2. Study Subjects

Potential blood donors had to fill out a written questionnaire and have a short-term health check carried out at the blood bank by a blood bank physician. To be allowed for blood donation, individuals had to meet the Saudi MOH’s donation eligibility requirements, which were assessed using the American Association of Blood Banks’ regulations and standards. All the study participants met the prerequisites for blood donation. Participants in this study were required to be free of COVID-19 symptoms and have never been infected with SARS-CoV-2. A blood donor was ruled out if he/she had been diagnosed with a disease or showed outward signs of infections such as cough, sore throat, or fever. Hepatitis B and C, as well as human immunodeficiency virus (HIV) 1 and 2, were confirmed to be absent in all the participants. After being approved for blood donation, all eligible blood donors were included in this study if they signed the informed consent form. All of the blood donations were completed at Makkah’s Central Blood Bank. 

### 2.3. Ethical Consideration

The research protocol has been reviewed and approved by the local ethics commissions. All procedures performed in this research were in accordance with ethical standards of the institutional and national research boards. The approval letter was issued from IRB-Makkah with the reference number (H-02-K-076-0121-456).

### 2.4. Participants’ Blood Samples

A total of 4368 residual serum samples, which remained after the routine screening of blood from eligible blood donors, obtained between 1 August and 31 December 2020, were screened for the presence of anti-SARS-CoV-2 IgG antibodies and SARS-CoV-2 RNA.

### 2.5. Enzyme-Linked Immunosorbent Assay (ELISA)

Serological detection of IgM and IgG could be a useful measure for identifying the prevalence and incidence of SARS-CoV-2 infections in the general population [22]. In this study, COVID-19-spike-specific IgG antibodies were detected by an ELISA kit according to the manufacturer’s instructions (Beijing BGI-GBI Biotech Co., Ltd., Shenzhen, China). Briefly, leave two wells for the negative control, one well for the positive control, and one well for the blank and add 100 μL of positive control or negative control in the designated wells without dilution. No liquid is added to the blank well. Add 100 μL of sample diluent buffer followed by 10 μL of the sample for testing to each of the other coated (with purified SARS-CoV-2 virus antigen) plate wells. Gently agitate the plate to fully mix the contents and seal the plate with a sealing paper. Incubate the plate at 37 °C for 30 min. Then, wash each well with 300 μL diluted washing buffer (PBST). Wait for 5 to 10 s, and discard the liquid contents. Repeat the washing step five times and pat the plate upside down on a tissue to dry. Add 100 μL of the enzyme solution (Horseradish peroxidase-labeled anti-human IgG antibody) to each well. Seal the plate with a sealing paper and incubate at 37 °C for 20 min. Repeat the washing step as above and then add 50 μL of Substrate A (Urea peroxide solution) and 50 μL of Substrate B (TMB solution) to each well. Mix thoroughly, seal the plate with a sealing paper, and incubate at 37 °C for 10 min shielded from light. Then, add 50 μL of the stop buffer to each well, and mix thoroughly. After quenching the reaction, place the plate in a microplate reader immediately to read the OD value at 450 nm (use the blank well to zero the reader). Double wavelength at 450 nm/620 nm was recommended to use. Quality control of each test dictate that the OD value of the positive control must be ≥0.50, and the OD value of the negative control must be ≤0.10. Otherwise, the test result is considered invalid. A negative test designate that a person has not mounted a sufficient immune response to SARS-CoV-2 and a positive one indicates that the person may have been exposed to SARS-CoV-2 and should be combined with clinical symptoms and other diagnostic results for further confirmation.

### 2.6. Results Calculation and Analysis

Cut-off value = 0.10 + mean OD value of negative control (Calculated as 0.05 if mean OD value of negative control is <0.05). If the OD value of a tested sample is greater than the cut-off value, the result is considered positive for IgG antibody against SARS-CoV-2. If the OD value of a tested sample is less than the cut off value, the result is considered negative for IgG antibody against SARS-CoV-2. If a sample OD value is close to the cut off value (OD value 0.12~0.18), it is recommended that the sample be re-tested. If the retested, result is greater than the cut off value, the sample is considered positive, otherwise, negative. Weakly positive samples were double checked with a different CE-certified test. We had 10 weak-positive samples, of which seven turned out as negative and three turned out as positive after repeated testing.

#### 2.6.1. RT-PCR

To confirm current or previous infection with SARS-CoV-2, real-time reverse-transcriptase polymerase chain reaction (RT-PCR) was used as described [23,24] to test for the presence of COVID-19 RNA among the IgG positive patients.

#### 2.6.2. Statistical Analyses

Statistical analyses were conducted using SPSS 24 (IBM SPSS Statistics for Windows, Version 24.0. IBM Corp., Armonk, NY, USA). The outcome variable was SARS-CoV-2 seropositivity, defined as an OD450 value of 0.30 or higher. Independent variables included age, nationality and blood group. Age was divided into the following two categories: up to 39 years and 40 years and above. Seropositivity outcomes were compared in the different age and blood groups. Statistical significance for both comparisons was assessed using the Chi-square test. Fisher’s exact test was used when comparing variables with values less than 5. An analysis of variance (ANOVA) was used to compare the IgG level means among age subgroups as well as the blood ABO groups.

## 3. Results

### 3.1. Prevalence of COVID-19 IgG, Sociodemographic and Clinical Characteristics of the Study Population

This study was conducted in the period from 1 August to 31 December of 2020 and included 4368 blood donors of whom 4339 were males (99.3%). According to the ELISA results, the participants were classified into COVID-19 IgG positive and IgG negative subjects represented by 2248 (51.46%) and 2120 (48.54%) subjects, respectively. As shown in Table 1, the mean OD (±SD) IgG level for all cases was 6.19 ± 6.85 with a range of 0.001–23.99. In this regard, the mean IgG OD was 11.96 ± 4.75 and 0.07 ± 0.15 among the IgG positive and IgG negative blood donors, respectively (*p* < 0.001). For all cases, the age range was 18–70 years with an overall average age of 32.92 ± 8.02 years with a significantly younger age in the IgG positive donors (*p* < 0.001). The percentage of COVID-19 IgG positive subjects was significantly higher among male donors (51.5%) when compared to 6.9% among females (*p* < 0.001; Fisher’s exact test). However, it should be taken in consideration that the female donors were only 29 subjects (0.7%). 

The distribution of COVID-19 IgG positive cases among different age groups is shown in Table 1 with the highest prevalence among those aging < 30 years with a significant variation among the different age groups (*p* < 0.001). In addition, the percentage of COVID-19 IgG positive donors was significantly higher among non-Saudi donors (58.9%) when compared to Saudi citizens (37.3%). In addition, there was a non-significant difference in the prevalence of COVID-19 IgG among blood donors with different blood groups (Table 1, *p* = 0.49). Significantly, the percentage of COVID-19 RNA positive cases (n = 473) determined by RT-PCR among the IgG positive donors was 21.04% (10.8% of the total subjects) with 1775 IgG positive donors being negative for SARS-CoV-2 RNA.

### 3.2. Assessment of COVID-19 IgG Positivity Risk with Age, Gender, Nationality and Blood Group

To assess the association risk of having COVID-19 IgG with different demographic characteristics, the blood donors were classified into two main age categories: those under 40 years or 40 years and above (n = 3486 and 882, respectively). In this regard, as shown in Table 2, the percentage of COVID-19 IgG positive subjects was 52.6 and 46.9 among those age groups, respectively (*p* = 0.001, χ^2^ = 9.07). In addition, the percentage of COVID-19 IgG positive donors was significantly higher (*p* < 0.001, Fisher’s Exact = 18.37) among males (51.8%) when compared to females with only two positive cases (6.9%) despite the low number of female donors (n = 29). In addition, the COVID-19 IgG positive cases were significantly (*p* < 0.001, χ^2^ = 184.2) higher among non-Saudi residents when compared to Saudi citizens (58.9% vs. 37.3%, respectively). On the other hand, there was no significant association between COVID-19 IgG positive cases and different blood groups with blood group “O” being the reference (*p* = 0.254, χ^2^ = 0.48, Table 2).

### 3.3. Assessment of COVID-19 RNA Positivity Risk with Age, Nationality and Blood Group within the COVID-19 IgG Positive Donors

The association risk of having COVID-19 RNA with different demographic characteristics within the COVID-19 IgG positive donors (n = 2248) was assessed. As shown in Table 3, the COVID-19 RNA positive cases were significantly (*p* = 0.047, χ^2^ = 3.95) different among blood donors who aged less than 40 years and those who are 40 years or older (20.3% and 24.8%, respectively). In addition, COVID-19 RNA positive cases were significantly (*p* = 0.001, χ^2^ = 304.5) higher among non-Saudi residents when compared to Saudi citizens (47.1% vs. 12.4%, respectively). On the other hand, there was no significant association between COVID-19 RNA positivity and different blood groups with blood group “O” being the reference (*p* = 0.329, χ^2^ = 0.245, Table 3). It should be noted that gender was not examined in this analysis due to the presence of only two COVID-19 RNA positive females.

### 3.4. Assessment of the Association of COVID-19 IgG Levels with COVID-19 RNA, Age, Nationality and Blood Group

The mean COVID-19 IgG levels was significantly (*p* = 0.001) higher among the COVID-19 RT-PCR-positive (n = 473) and negative (n = 1775) blood donors measuring 12.99 ± 4.68 and 11.69 ± 4.73, respectively. In addition, COVID-19 IgG levels were significantly (*p* = 0.041) higher among non-Saudi residents when compared to Saudi citizens (12.36 ± 5.07 Vs 11.87 ± 4.67, respectively). However, there was no association between COVID-19 IgG levels and the age or blood group of the study participants (*p* > 0.05, Table 4). 

### 3.5. Comparison between the Mean IgG Level among the IgG Positive Blood Donors from Different Age Groups

Analysis of the COVID-19 IgG levels among the different age groups is shown in Table 5. As shown, there was a significantly higher levels of COVID-19 IgG among the older blood donors (45–49 years and 50 years or older with a *p* value of 0.016 and 0.024, respectively) as compared to the rest of the age groups. ANOVA testing revealed an F of 3.23 and a *p* value of 0.018 indicating significant variation in the COVID-19 IgG levels among the different age groups.

### 3.6. Comparison between the Mean IgG Level among the IgG Positive Blood Donors with Blood Groups

Analysis of the COVID-19 IgG levels among the different blood groups is shown in Table 6. As shown, there was a non-significant variation in the COVID-19 IgG levels among the blood donors with different blood groups (*p* > 0.05). ANOVA testing revealed an F of 1.38 and a *p* value of 0.248 indicating a non-significant variation in the COVID-19 IgG levels among the blood donors with different blood groups.

## 4. Discussion

This study examined the presence of SARS-CoV-2 spike-specific IgG among 4368 asymptomatic blood donors in Makkah region between 1 August and 31 December 2020. COVID-19 IgG was detected among 2248 subjects (51.5%) while COVID-19 RNA was detected among 473 (10.8%) subjects showing a high prevalence of COVID-19 IgG and RNA among asymptomatic blood donors in Makkah, Saudi Arabia. This study shows that the frequency of IgG was significantly higher among males and non-Saudi residents with no significant variation in IgG positivity among those with different blood groups. In addition, COVID-19 RNA frequency was significantly higher among those below 40 years old and non-Saudi residents. The COVID-19 IgG levels were significantly higher among the RNA-positive donors, and non-Saudi residents with no variations with age or blood group. Several aspects of these data deserve further discussion.

The first half of this study period represents the last period of the first peak of COVID-19 in Saudi Arabia (April to October 2020) while the second half represents the beginning of a period where the cases were in the steady state (https://graphics.reuters.com/world-coronavirus-tracker-and-maps/countries-and-territories/saudi-arabia/, accessed on 17 March 2022). It is known that SARS-CoV-2 can cause infections that range from asymptomatic to severe infections and even death [25,26,27]. This study shows a high prevalence of COVID-19-spike specific IgG (51.5%) and 10.8% having COVID-19 RNA suggesting a very high exposure to this respiratory virus without symptomatic manifestation. It emphasizes the importance of serological and molecular surveillance to determine the extent of viral transmission among the general population and stresses the importance of imposing different infection control measures to minimize viral spread. This could have important implications in determining herd immunity among the general population.

A lot of worldwide studies have examined the seroprevalence of COVID-19 IgG among asymptomatic subjects with inconsistent results. In this regard, a study in California, USA showed a 2.8% seroprevalence among healthy individuals [28]. Another study among 3068 asymptomatic Japanese subjects showed an overall COVID-19 seroprevalence of 17.9% [29]. Several low COVID-19 seroprevalence rates were reported including 0% in Jordan [30], 0.1% in California, USA [31], 1 to 23% among 177,919 subjects in 52 USA jurisdictions [32], 1.9% in Denmark [33], 2.7% in blood donors in the Netherlands [34], 3.3% among blood donors in Brazil [35], 8% among 28,503 renal dialysis patients in USA [36], and 17.1% in Iran [37]. Higher COVID-19 seroprevalences were also reported reaching 22.6–23% in Italy [38,39], 22.7% in New York City [40], 31.6% and 38% among HCWs in two different studies in Madrid, Spain [41,42], 44% in the Brazilian Amazon [43], and 65% among 60 seniors in Germany [44]. In Saudi Arabia, a small number of COVID-19 IgG seroprevalence studies were conducted with varying frequencies ranging from 0% to 32.2% [11,12,13,14,15,45]. Three of these studies were among blood donors during the early stages of the pandemic (January to May, May, and June 2020) [12,13,14], while one looked at healthcare workers (HCWs) in May 2020 under the umbrella of the Saudi Ministry of Health and the Saudi Center for Disease Control [11]. The later study [11] examined 12621 HCWs in 85 different hospitals with an overall 2.36% IgG prevalence [11]. In the same theme, two reports (one included 24 blood banks, n = 837 [14], and one from the western region of Saudi Arabia, n = 956 [13]), showed that the seroprevalence was 1.4% and 0%, among blood donors and HCWs, respectively. Another report from Al-Madinah (n = 1212) showed a much higher prevalence (19.3%) among asymptomatic blood donors [12]. Our study looked at SARS-CoV-2-specific IgG seroprevalence among 4368 blood donors in Makkah after the first COVID-19 peak and before the introduction of the vaccine and showed a much higher IgG prevalence than the above-mentioned reports, reaching 51.5% prevalence of IgG and 10.8% prevalence of RNA. This suggests the existence of a very high exposure to the virus without symptomatic manifestations among the healthy blood donors enrolled in this study. To this end, the controversy in prevalence rates can be attributed to several factors including the study period, population characteristics, methods of detection of the antibodies, and the study region or country.

In this study, COVID-19 IgG seroprevalence and levels were significantly higher among donors with an age of 40 years or more when compared with those aging 18–39 years. In addition, the IgG levels were higher among those aging 45–49 and ≥50 years as compared to younger blood donors. In this regard, a study from Al-Madinah [12] and all over Saudi Arabia [14] among blood donors did not find any significant difference in the seropositivity for COVID-19 IgG with different age. Another study in Spain, additionally, reported no difference in COVID-19 IgG seroprevalence among subjects with different ages [41].

This study showed that there were no significant differences in the seroprevalence of COVID-19 IgG or even the IgG levels among blood donors with different blood groups. This was, additionally, true for COVID-19 RNA. In this regard, the study performed in Al-Madinah [12] found a significantly higher prevalence of COVID-19 antibodies among the blood donors (n = 1212) with blood group A, which contradicts our findings. It should be noted that the number of participants in this study is almost four-fold that examined in Al-Madinah. In the same argument, another meta-analysis reported a higher seroprevalence of SARS-CoV-2 among blood group A subjects [46].

The higher prevalence of COVID-19 infection among non-Saudi residents as compared to Saudi citizens found in this study have been previously described [47]. The higher prevalence of IgG, its level and COVID-19 RNA among the non-Saudi residents may be attributed to their low level of education, sociodemographic characteristics [48] and the degree of adherence to the preventive measure imposed by the local authorities. Other reports, additionally, showed this trend among blood donors in the kingdom [14] and among individuals with different ethnicities in the United States [48]. Importantly, the sensitivity and specificity of the COVID-19-IgG kit used in this study are 93.78 and 97.12, respectively. In addition, the PCR technique is highly specific, and the likelihood of contamination during the assay is, additionally, minimal. In addition, it is known that COVID-19 is a respiratory disease that is asymptomatic in a high percentage of subjects. These data suggest a minimal error; if any; on the overall prevalence of IgG and RNA in this study. 

Although this cross-sectional study has several strengths including the large number of participants, and testing COVID-19 RNA among the seropositive subjects, it has some limitations. First, the limitation of cross-sectional studies. Second, we measured only IgG but not IgM, which indicates recent exposure to the virus. Third, the number of females in this study was low (n = 29). Therefore, sex differences should be interpreted cautiously. Finally, examining the duration and protective nature of the immune responses reported herein among the asymptomatic blood donors against COVID-19 is crucial to prevent further spread of the virus, which was not examined in this study.

In summary, this study showed a very high prevalence of COVID-19 IgG and RNA among asymptomatic blood donors in Makkah, indicating a high exposure rate of the general population of Saudi Arabia, particularly foreign residents. In conclusion, this study sheds the light on the spread on COVID-19 among apparently healthy individuals in Saudi Arabia at the beginning of the pandemic and could help in designing various control measures to minimize viral spread.

## Figures and Tables

**Table 1 vaccines-10-01279-t001:** Frequency distribution and descriptive statistics of sociodemographic and clinical characteristics of all the blood donors who participated in the study.

		Total Participants	IgG Positive	IgG Negative	*p*-Value
		N = 4368	n = 2248	n = 2120	
**Age**	**Mean ± SD**	32.92 + 8.02	31.96 + 7.89	33.95 + 8.03	*p* < 0.001
IgG level ^a^	Mean ± SD	6.19 + 6.85	11.96 + 4.75	0.07 + 0.15	*p* < 0.001
Sex	Male Female	4339 (99.3%) 29 (0.7%)	2246 (51.8%) 2 (6.9%)	2093 (48.2%) 27 (93.1%)	*p* < 0.001
Age/years ^b^	18–24	625 (14.3%)	390 (62.4%)	235 (37.6%)	*p* < 0.001
	25–29	987 (22.6%)	586 (59.4%)	401 (40.6%)	
	30–34	1024 (23.4%)	487 (47.6%)	537 (52.4%)	
	35–39	850 (19.5%)	371 (43.6%)	479 (56.4%)	
	40–44	521 (11.9%)	252 (48.4%)	269 (51.6%)	
	45–49	196 (4.5%)	96 (49.0%)	100 (51.0%)	
	≥50	165 (3.8%)	66 (40.0%)	99 (24.8%)	
Nationality	Saudi Non-Saudi	1504 (34.4%) 2864 (65.6%)	561 (37.3%) 1687 (58.9%)	943 (62.7%) 1177 (41.1%)	*p* < 0.001
Blood Group	A	1172 (26.8%)	610 (52.0%)	562 (48.0%)	*p* < 0.490
	B	982 (22.5%)	510 (51.9%)	472 (48.1%)	
	AB	260 (6.0%)	143 (55.0%)	117 (45.0%)	
	O	1954 (44.7)	985 (50.4%)	969 (49.6%)	
PCR results	Positive Negative	473 (10.8%) 3895 (89.2%)	473 (100.0%) 1775 (45.6%)	Not applicable	*p* < 0.001

^a^ For all cases, the level of IgG was expressed as OD (Mean ± SD 6.19 ± 6.85, and Range = 0.001–23.99). ^b^ For all cases, age range was 18–70 years.

**Table 2 vaccines-10-01279-t002:** Risk of having positive IgG by age, sex, nationality and blood group.

	Variable	IgG Positive n = 2248	IgG Negative n = 2120	OR (95% CI)	Chi^2^ *p*-Value
Age/years	Up to 39, n = 3486	1834 (52.6%)	1652 (47.4%)	Reference	X^2^ = 9.07 *p* = 0.001
	40 years or more n = 882	414 (46.9%)	468 (53.1%)	1.12 (1.04–1.21)	
Sex	Female, n = 29	2 (6.9%)	27 (93.1%)	Reference	(Fisher’s = 18.37) *p* < 0.001 ^a^
	Male, n = 4339	2246 (51.8%)	2093 (48.2%)	7.5 (1.97–28.59)	
Nationality	Saudi, n = 1504	561 (37.3%)	943 (62.7%)	Reference	X^2^ = 184.2 *p* = 0.001
	Non-Saudi, n = 2864	1687 (58.9%)	1177 (41.1%)	2.41 (2.12–2.74)	
Blood Group	O, n = 1936	985 (50.9%)	951 (49.1%)	Reference	X^2^ = 0.480 *p* = 0.254
	Others (A, B, AB) n = 2432	1263 (51.9%)	1169 (48.1%)	1.02 (0.93–1.18)	

^a^ Fisher’s exact test.

**Table 3 vaccines-10-01279-t003:** PCR results association with age, nationality and blood groups within the positive IgG cases.

		Number	COVID-19 PCR −Ve	COVID-19 PCR +Ve	Chi Squared *p*-Value
Age/years	Up to 39 years	1834	1463 (79.7%)	371 (20.3%)	Chi = 3.95 *p* = 0.047
	40 years and more	414	312 (75.2%)	102 (24.8%)	
Nationality	Saudi	561	297 (52.9%)	264 (47.1%)	Chi = 304.5 *p* = 0.001
	Non-Saudi	1687	1478 (87.6%)	209 (12.4%)	
Blood Group	O	985	773 (78.5%)	212 (21.5%)	Chi = 0.245 *p* = 0.329
	Others (A, B, AB)	1263	1002 (79.3%)	261 (20.7%)	

**Table 4 vaccines-10-01279-t004:** The level of IgG divided by COVID-19 PCR results, age, nationality and ABO group.

Parameter	IgG OD Mean ± SD	*p*-Value
PCR COVID-19 Negative (n = 1775)	11.69 ± 4.73	0.001
PCR COVID-19 Positive (n = 473)	12.99 ± 4.68	
Age up to 39 years old (n = 1834)	11.87 ± 4.67	0.063
Age of 40 years and above (n = 414)	12.36 ± 5.07	
Saudi citizens (n = 561)	11.61 ± 4.89	0.041
Non-Saudi residents (n = 1687)	12.08 ± 4.69	
O group blood donors (n = 985)	11.79 ± 4.81	0.129
A, B or AB groups blood donors (n = 1263)	12.09 ± 4.70	

**Table 5 vaccines-10-01279-t005:** Comparison between the mean IgG level among the IgG positive blood donors from different age groups.

Age Group	N = 2248 N (%)	Mean IgG Units/mL	SD	SE	*p*-Value ^a^
18–24 y	390 (17.3%)	11.74	4.34	0.296	0.018
25–29 y	586 (26.1%)	11.73	4.66	0.292	
30–34 y	487 (21.7%)	12.20	4.74	0.215	
35–39 y	371 (16.5%)	11.81	4.92	0.265	
40–44 y	252 (11.2%)	11.71	5.16	0.327	
45–49 y	96 (4.3%)	13.54	4.62	0.472	
≥50 y	66 (2.9%)	13.11	4.86	0.598	

^a^ ANOVA test: F = 3.23.

**Table 6 vaccines-10-01279-t006:** Comparison between the mean IgG level among blood donors of different ABO blood groups.

ABO Group	Number = 2248 N (%)	Mean IgG OD	SD	SE	*p*-Value ^a^
O	985 (43.8%)	11.79	4.81	0.153	0.248
A	610 (27.1%)	12.19	4.86	0.197	
B	510 (22.7%)	12.12	4.52	0.201	
AB	143 (6.4%)	11.56	4.58	0.383	

^a^ ANOVA test: F = 1.38.

## Data Availability

All data pertinent to this study are included herein.

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
