# Peer review of "High Seroprevalence of SARS-CoV-2 IgG and RNA among Asymptomatic Blood Donors in Makkah Region, Saudi Arabia"

_vaccines, 2022, doi:10.3390/vaccines10081279_

Round 1

Reviewer 1 Report

The manuscript titled High Seroprevalence of SARS-CoV-2 IgG and RNA Among Asymptomatic Blood Donors in Makkah Region, Saudi Arabia by Alzabeedi et al. examined the presence of SARS-CoV-2 spike-specific immunoglobulin G (IgG) among 4368 asymptomatic blood donors in Makkah region and found a very high prevalence of COVID-19 IgG and COVID-19 RNA among asymptomatic blood donors in Makkah, Saudi Arabia indicating a high exposure rate of the general population to COVID-19; particularly foreign residents. The study is well described. There are only several minor suggestions, which may help to improve this manuscript.

  1. Check the sample size for Sex in table 2, which likely has typos.

  2. The female sample only has 2 IgG Positive cases, the chi square test is not appropriate, consider using the fisher’s exact test instead of the chi square test.

  3. 99.3% of the subjects are male, why not consider excluding females from this study? There are not enough female samples to claim the sex difference, consider dropping the sex variable?

  4. In Table 5, the notes stated ANOVA test: F = 3.23 and p=0.018,  the p value column showed values from the post hoc tests for each group compared with the reference group?  Did you adjust the post hoc p values for multiple comparisons? Similarly in Table 6, the ANOVA test showed no significant effect of the ABO group,  no need to compare  across ABO  group?

  5. The notes for Table 1 need to be reformatted.

Author Response

thank the reviewer for the comment. We followed this comment by using the ANGeneral comment: The manuscript titled “High Seroprevalence of SARS-CoV-2 IgG and RNA Among Asymptomatic Blood Donors in Makkah Region, Saudi Arabia” by Alzabeedi et al. examined the presence of SARS-CoV-2 spike-specific immunoglobulin G (IgG) among 4368 asymptomatic blood donors in Makkah region and found a very high prevalence of COVID-19 IgG and COVID-19 RNA among asymptomatic blood donors in Makkah, Saudi Arabia indicating a high exposure rate of the general population to COVID-19; particularly foreign residents. The study is well described. There are only several minor suggestions, which may help to improve this manuscript.

Response to general comment: We thank the reviewer for commending the manuscript.

Comment #1: Check the sample size for Sex in table 2, which likely has typos.

Response to Comment #1: Thanks for the reviewer for the thorough revision. Yes, there was a typing error for the numbers mentioned in Table 2 and it is now corrected.

Comment #2: The female sample only has 2 IgG Positive cases, the chi square test is not appropriate, consider using the fisher’s exact test instead of the chi square test.

Response to Comment #2: Thanks for the comment. Fisher's exact test was calculated and mentioned in the Table and its corresponding text.

Comment #3: 99.3% of the subjects are male, why not consider excluding females from this study? There are not enough female samples to claim the sex difference, consider dropping the sex variable?

Response to Comment #3: We agree with the reviewer that the number of female blood donors is very small compared to that of the male donors. However, the culture and practice of blood donation among Saudi females was very infrequent. Nowadays, it is starting to grow, and the culture is changing, albeit very slowly. We prefer to keep the females in this study as evidence of this change in culture. This was indicated in the study limitations at the end of the discussion. We, also, added the following statement in the study limitations section “Therefore, sex differences should be interpreted cautiously”.

Comment #4: In Table 5, the notes stated ANOVA test: F = 3.23 and p=0.018,  the p value column showed values from the post hoc tests for each group compared with the reference group?  Did you adjust the post hoc p values for multiple comparisons? Similarly in Table 6, the ANOVA test showed no significant effect of the ABO group, no need to compare  across ABO  group?

Response to Comment #4: We OVA p-value without presenting the post hoc values  for Age groups (Table 5) and ABO blood groups (Table 6).

Comment #5: The notes for Table 1 need to be reformatted.

Response to Comment #5: The footnote for Table 1 was amended as recommended by the reviewer.

Reviewer 2 Report

The manuscript by Alzabeedi et al examines the prevalence of spike-specific IgG and COVID-19 RNA in blood donors in Saudi Arabia autumn 2020. They suggest a high prevalence of IgG (50%) among asymptomatic blood donors and also a high prevalence of PCR positivity (10%).

My main concern is if false positive results may be present for IgG and PCR positive donors, since the frequency of positive donors is so high. If the authors have samples collected before the pandemic if would be valuable to analyse those samples to make sure no positive results are detected.

Author Response

The manuscript by Alzabeedi et al examines the prevalence of spike-specific IgG and COVID-19 RNA in blood donors in Saudi Arabia autumn 2020. They suggest a high prevalence of IgG (50%) among asymptomatic blood donors and also a high prevalence of PCR positivity (10%). My main concern is if false positive results may be present for IgG and PCR positive donors, since the frequency of positive donors is so high. If the authors have samples collected before the pandemic if would be valuable to analyse those samples to make sure no positive results are detected.

Response to general comment: The sensitivity and specificity of the COVID-19-IgG kit used in this study are 93.78 and 97.12, respectively. The IgG kit suggests that the main cause of false positive results is insufficient plate washing or other mistakes that may lead to false positive- or negative- results including kit usage after expiration; the pipettors used are inaccurate; the operating temperature is too low (<10°C); or the procedures outlined in this protocol were insufficiently adhered to. Also, the PCR molecular technique is highly specific, and the likelihood of contamination during the assay is also minimal. In addition, it is known that COVID-19 is a respiratory disease that is asymptomatic in a high percentage of subjects. These data suggest a minimal error; if any; on the overall prevalence of IgG and RNA in this study. This meaning was amended in the discussion section. Furthermore, for this study, we used residual serum samples obtained between August 1st and December 31st, 2020 (which remained after the routine screening of blood from eligible blood donors). We do not have samples before the pandemic as they are routinely discarded and no regular blood donation from the same donors.

Reviewer 3 Report

The manuscript: vaccines-1798268, that described the high Seroprevalence of SARS-CoV-2 IgG and RNA Among Asymptomatic Blood Donors in Makkah Region, Saudi Arabia by Kamal H. Alzabeedi. This is interesting but lack of information, need to update references. The method should be improved as it has flaws. Overall, language should be edited. So, I recommend major revisions.

Author Response

General comment: The manuscript: vaccines-1798268, that described the high Seroprevalence of SARS-CoV-2 IgG and RNA Among Asymptomatic Blood Donors in Makkah Region, Saudi Arabia by Kamal H. Alzabeedi. This is interesting but lack of information, need to update references. The method should be improved as it has flaws. Overall, language should be edited. So, I recommend major revisions.

Response to general comment:  As recommended by the reviewer, the methodology, references and English were revised. Two new references were cited.

Reviewer 4 Report

There are several inconsistency and errors throughout the manuscript as shown below.

Introduction section

1. The authors mentioned the imported cases from Iran in March, 2020, then the virus slowly transmitted in the community (Line 60-67). However, the kingdom experienced a peak from June to August, 2020. Is it related to the imported cases? Then Why the lockdown was lifted in June 21 (Line 71) during the peak of cases? The authors claimed this lift of lockdown was owing to the excellent response of the authority even though 6445 patients died in COVID-19 by Feb 2021.

2.   Line 81, the sentence "20.1% of infected patients suffering from underlying comorbidity" required clarification. Suggested to "Should be infected patients with underlying comorbidity suffering from severe diseases".

3. Line 83, RT-PCR is capable of detecting dormant viral infection, such as HBV or herpesvirus. The sentence requires clarification.

4. Line 108-109, the authors claimed that no one looked at the seroprevalence after the peak. One reference published by Alhazmi et al in Int J Environ Res Public Health2021 Nov 26;18(23):12451 looked at the seroprevalence from Aug to December, 2020. The exact duration of the current study.

5. Line 120-121, the blood samples were collected not during the peak, but after as described in Materials/Methods section (Line 153)

Materials and Methods section

1. The authors didn't describe how to measure IgG level as IU. Does it be measured as Antigen-specific IgG or general IgG? In Table 1, even the IgG negative results has the IgG level measured. How did it be derived? Since the IgG levels are compared throughout the results section, the details need to be provided.

2. Line 185-187, if re-tested still at border line, what the authors would do? treated as weak positive samples and re-tested by a different certified assay? The details of which one and how many samples should be described.

3. Which results used ANOVA test? If no result used ANOVA, the sentence should be deleted.

Results section

1. Table 1, since no IgG negative samples were evaluated by RT-PCR, it should be "non-applicable". Also, under the Table, those descriptions were for footnote or ?

2. Many errors in Table 2 and the corresponding section 3.2. IgG positve n=8422 is wrong. OR is wrongly calculated with the wrong reference. As for gender, since only 2 in female, the test should use Fisher's exact test.

3. Table 3 is correct but the corresponding section 3.3 is wrongly described regarding the percentage of PCR positive samples.

Author Response

General comment: There are several inconsistency and errors throughout the manuscript as shown below.

Response to general comment: We thank the reviewer for the comment that improved the overall quality of the manuscript, and the inconsistences were fixed in the revised version of the manuscript.

Introduction section

Comment #1: The authors mentioned the imported cases from Iran in March, 2020, then the virus slowly transmitted in the community (Line 60-67). However, the kingdom experienced a peak from June to August, 2020. Is it related to the imported cases? Then Why the lockdown was lifted in June 21 (Line 71) during the peak of cases? The authors claimed this lift of lockdown was owing to the excellent response of the authority even though 6445 patients died in COVID-19 by Feb 2021.

Response to Comment #1: Although the spread of the virus may not be related to imported cases (as detailed in line 63-68), this is the first documented case in Saudi Arabia. Also, death cases are spread all over the world regardless of the measures implemented. In addition, the lockdown was very strict, and most of the dead cases were belonging to patients with comorbidities e.g., chronic diseases and old ages.

Comment #2: Line 81, the sentence "20.1% of infected patients suffering from underlying comorbidity" required clarification. Suggested to "Should be infected patients with underlying comorbidity suffering from severe diseases".

Response to Comment #2: the sentence was clarified as this sentence was a citation from reference #2.

Comment #3: Line 83, RT-PCR is capable of detecting dormant viral infection, such as HBV or herpesvirus. The sentence requires clarification.

Response to Comment #3: The sentence was clarified as suggested by the reviewer.

Comment #4: Line 108-109, the authors claimed that no one looked at the seroprevalence after the peak. One reference published by Alhazmi et al in Int J Environ Res Public Health. 2021 Nov 26;18(23):12451 looked at the seroprevalence from Aug to December, 2020. The exact duration of the current study.

Response to Comment #4: We thank the reviewer for the comment and agree that there are several prevalence studies conducted in Saudi Arabia as shown in the table below )from Alhazmi et al., 2021) and the bibliography of the manuscript. However, there was only one study from Makkah on 204 healthcare workers. These studies were cited in the manuscript and the suggested paper (Alhazmi et al., 2021) was also cited in the revised manuscript. Also, the sentence was amended.

Comment #5: Line 120-121, the blood samples were collected not during the peak, but after as described in Materials/Methods section (Line 153)

Response to Comment #5: We thank the reviewer for the catch. This sentence was amended to indicate “after the peak” as the collection period.

Materials and Methods section

Comment #6: The authors didn't describe how to measure IgG level as IU. Does it be measured as Antigen-specific IgG or general IgG? In Table 1, even the IgG negative results has the IgG level measured. How did it be derived? Since the IgG levels are compared throughout the results section, the details need to be provided.

Response to Comment #6: We thank the reviewer for the comment. We measured COVID-19-spike-specific IgG antibodies as indicated in section 2.5 of the Materials and Methods section. The level of IgG was expressed as OD as detailed section 2.5. The IgG negative cases have an OD measurement below the cut off value for a positive test. Anyone can refer to the kit insert for the detailed information/ procedures. The units/ml in Table 1 footnote and some other sections was a typing error.

Comment #7: Line 185-187, if re-tested still at border line, what the authors would do? treated as weak positive samples and re-tested by a different certified assay? The details of which one and how many samples should be described.

Response to Comment #7: As indicated at the end of section 2.6, weakly positive samples were checked with a different CE-certified test. We have 10 weak-positive samples of which seven were negative and three were positive after repeated testing. This was added at the end of section 2.6.

Comment #8: Which results used ANOVA test? If no result used ANOVA, the sentence should be deleted.

Response to Comment #8: Regarding ANOVA test that was presented in the methodology section, we made it clear in section 2.6.2. Also, it is highlighted under Tables 5 and 6, where this ANOVA test was applied.

Results section

Comment #9: Table 1, since no IgG negative samples were evaluated by RT-PCR, it should be "non-applicable". Also, under the Table, those descriptions were for footnote or ?

Response to Comment #49: We changed this to “non-applicable” and the footnote was amended using reference letters within the table.

Comment #10: Many errors in Table 2 and the corresponding section 3.2. IgG positive n=8422 is wrong. OR is wrongly calculated with the wrong reference. As for gender, since only 2 in female, the test should use Fisher's exact test.

Response to Comment #10: Thank you very much dear reviewer for re-checking the numbers of Table 2 and its corresponding section. We revised and corrected the total number of IgG positive cases n=2248  and we used the Fisher's exact test for comparisons because of the small number in the cell that contains only two female IgG positive cases.

Comment #11: Table 3 is correct but the corresponding section 3.3 is wrongly described regarding the percentage of PCR positive samples.

Response to Comment #11: Thanks a lot for the comment. The percentage of PCR positive samples with age groups was corrected in the corresponding section of Table 3 (section  3.3).

Round 2

Reviewer 2 Report

I am still not convinced by the specificity of the IgG assay in the hands of the researchers. Hence, my suggestion would be to find serum samples from any lab from before the pandemic and run the assay on that serum. 

Author Response

Comment #1: I am still not convinced by the specificity of the IgG assay in the hands of the researchers. Hence, my suggestion would be to find serum samples from any lab from before the pandemic and run the assay on that serum. 

Response to comment #1:

In the current study, our main objective was to determine the seroprevalence of COVID-19 IgG among the enrolled blood donors. Therefore, studying the sensitivity and specificity of the kit used in the study is beyond the scope of this manuscript. Also, we would like to clarify that the IgG kit used in the study was chosen and decided by the Saudi Governmental Laboratories that was certified and approved by the Ministry of Health in the Kingdom. Importantly, we addressed the concern of the reviewer in the revised version (line 373-378), where we clearly stated the specificity and sensitivity of the assay. Additionally, a similar study by Alhabbab et al., that enrolled only 693 healthcare workers in Jeddah (cited in the manuscript), also, showed a high seroprevalence rate reaching 32.2%.

Reviewer 3 Report

More patients could be evaulated..

Author Response

General comment #1: More patients could be evaluated.

Response to comment #1: We thank the reviewer for the comment. However, we would like to emphasize that one of the strengths of our study is the enrollment of the largest number of participants (4368 blood donors) compared to the other seroprevalence studies that were performed in KSA. We presented this information in the discussion section (line 379). Here are all the five studies that were conducted in Saudi Arabia since the beginning of the COVID-19 pandemic and their number of enrollees (all cited in the manuscript):

  1. Alandjiany et al. enrolled 956 blood donors in Jeddah,
  2. Mahallawi et al. enrolled 1212 blood donors in Al Madinah,
  3. Ahmed et.al. enrolled 204 heath care workers in Makkah,
  4. Alhabbab et. al. enrolled 693 heath care workers in Jeddah, and
  5. Alhazmi et. al. enrolled 594 individuals from the general population with seroprevalence 26% in Jazan, KSA.

From the above data, our enrollees (4368 blood donors) represent the largest number of enrollees among all the seroprevalence studies performed in Saudi Arabia reaching almost four-fold the number of enrollees in the largest study (#2).